# Current Opinions about the Use of Duloxetine: Results from a Survey Aimed at Psychiatrists

**DOI:** 10.3390/brainsci13020333

**Published:** 2023-02-15

**Authors:** M. A. Alvarez-Mon, Cielo García-Montero, Oscar Fraile-Martinez, Javier Quintero, Sonia Fernandez-Rojo, Fernando Mora, Luis Gutiérrez-Rojas, Rosa M. Molina-Ruiz, Guillermo Lahera, Melchor Álvarez-Mon, Miguel A. Ortega

**Affiliations:** 1Department of Psychiatry and Mental Health, Hospital Universitario Infanta Leonor, 28031 Madrid, Spain; 2Department of Medicine and Medical Specialities, Faculty of Medicine and Health Sciences, University of Alcalá, 28801 Alcalá de Henares, Spain; 3Ramón y Cajal Institute of Sanitary Research (IRYCIS), 28034 Madrid, Spain; 4Department of Legal Medicine and Psychiatry, Complutense University, 28040 Madrid, Spain; 5Department of Psychiatry, University of Granada, 18016 Granada, Spain; 6Psychiatry Service, San Cecilio University Hospital, 18016 Granada, Spain; 7Department of Psychiatry and Mental Health, Hospital Universitario Clínico San Carlos, 28040 Madrid, Spain; 8Department of Psychiatry, University Hospital Príncipe de Asturias, 28805 Alcalá de Henares, Spain; 9Mental Health Networking Biomedical Research Centre (CIBERSAM), 28029 Madrid, Spain; 10Immune System Diseases-Rheumatology and Internal Medicine Service, University Hospital Príncipe de Asturias, CIBEREHD, 28806 Alcalá de Henares, Spain

**Keywords:** duloxetine, major depressive disorder (MDD), real world data, serotonin and norepinephrine reuptake inhibitors (SNRIs)

## Abstract

Major depressive disorder (MDD) is a complex psychiatric disorder that, presented alone or with other comorbidities, requires different adjustments of antidepressant treatments. Some investigations have demonstrated that psychoactive drugs, such as serotonin and norepinephrine reuptake inhibitors (SNRIs), can exert more effective and faster antidepressant effects than other common medications used, such as serotonin selective reuptake inhibitors (SSRIs), although these differences are still controversial. During the last five years, the SNRI duloxetine has shown favorable results in clinical practice for the treatment of MDD, anxiety, and fibromyalgia. Through an online self-completed survey, in the present article, we collected information from 163 psychiatrists regarding the use of duloxetine and its comparison with other psychiatric drugs, concerning psychiatrists’ knowledge and experience, as well as patients’ preferences, symptoms, and well-being. We discussed and contrasted physicians’ reports and the scientific literature, finding satisfactory concordances, and finally concluded that there is agreement regarding the use of duloxetine, not only due to its tolerability and effectiveness but also due to the wide variety of situations in which it can be used (e.g., somatic symptoms in fibromyalgia, diabetes) as it relieves neuropathic pain as well.

## 1. Introduction

Major depressive disorder (MDD) is a growing and debilitating condition affecting among 2 to 21% of people with an estimated lifetime prevalence, and representing the leading cause of disability worldwide [1,2]. Clinically, MDD is characterized by the co-occurrence of either depressed mood or anhedonia (core symptoms) with at least four somatic and non-somatic items with detrimental consequences for the individual (i.e., sleep alterations, appetite changes, pain, feeling of worthlessness, suicidal ideation, and so on) [3]. The pathobiological basis of MDD remains poorly understood. Some previous works describe MDD as the result of a complex interplay between genetic and environmental factors, with a major role of changes in neurotransmission, altered neuroendocrine axis, neuroinflammation, epigenetic alterations, lifestyle behaviors, systemic inflammation, and changes in the microbiota-gut-brain axis [4,5,6,7,8]. Furthermore, MDD represents a huge and growing burden for society and healthcare systems. Apart from the direct costs derived from medical care, indirect costs related to a lack of productivity or work absence represent an important point to consider in MDD management [9]. In Spain alone, the prevalence of MDD was 4.73%, the incidence rates between 2015 and 2017 were 7.12, 7.35, and 8.02 per 1000 person/year respectively, and the estimated societal cost of MDD in the Spanish adult population is of € 6145, with a mean cost of € 3402 per patient, mostly derived from indirect rather than direct costs [10]. Hence, improving the clinical management of these patients is imperative, not only to improve their quality of life, but also to diminish the socioeconomic burden related to MDD.

The use of antidepressants is commonly recommended in the clinical management of MDD patients, which may be used alone or in combination with non-medical treatments such as physical activity, psychotherapy, or psychoeducation [11]. The type and quantities of antidepressants are gradually increasing, but the efficacy and safety of first line and emerging drugs varies greatly between studies. Furthermore, ascertaining what antidepressant will benefit each patient the most is a major challenge for psychiatrists, showing the need for work aiming to address this important issue [12]. Duloxetine is an antidepressant medication used in the clinical management of psychiatric and non-psychiatric conditions such as MDD, generalized anxiety disorder (GAD), fibromyalgia, diabetic peripheral neuropathy, and chronic musculoskeletal pain [13]. Pharmacologically, duloxetine belongs to the serotonin (5-HT) and noradrenaline (NA) reuptake inhibitors (SNRIs). These drugs impede the reuptake of both 5-HT and NA with different selectivity for each neurotransmitter [14]. Apart from duloxetine, other SNRIs are venlafaxine, milnacipran, desvenlafaxine, or levomilnacipran. It is known that each drug harbors several pharmacological differences, fostering a unique identity [15]. For instance, it is well established that duloxetine exerts a 10-fold higher selectivity for serotonin reuptake inhibition compared with norepinephrine reuptake inhibition [14]. In other words, reuptake inhibition seems to be asymmetrical after duloxetine administration, with an initial influence on serotonin followed by a modulation of norepinephrine and a slight modulation of other monoamines, such as dopamine [15].

According to the literature, duloxetine is one of the first-line antidepressants used globally [16]. In this sense, considering the experience and professional opinion shared by clinicians is essential to verify the use and applications of different antidepressant alternatives in the field of MDD [17,18]. Indeed, a growing body of evidence supports the relevance of obtaining field data for generating real-world evidence for designing and conducting confirmatory trials and answering questions that may not be addressed otherwise [19]. These types of studies can be particularly important in the area of mental health disorders, as they have the potential to improve the safety, quality, and effectiveness of care for these patients [20,21,22].

Thus, the objective of the present work is to collect information regarding the use of duloxetine in comparison with other psychiatric drugs and concerning psychiatrists’ knowledge and experience, as well as patients’ preferences, symptoms, and well-being. With this goal in mind, an online questionnaire was created and sent to 163 psychiatrists from different regions of Spain. We aimed to obtain real-world evidence of psychiatrists’ opinions.

## 2. Materials and Methods

This article collects the information from an exploratory online self-completed survey aimed at psychiatrists from different hospitals and mental health centers in Spain. The questionnaire consisted of 20 questions (4 of which were open-ended), designed by the authors and distributed by Adamed Laboratories (Madrid, Spain). The questions from our survey are presented in Table 1. A total of 163 professionals filled out the questionnaire between 15th May and 22nd September 2022. The inclusion criteria to be able to participate in the survey were: (1) being a psychiatrist; (2) having more than 5 years of work experience; and (3) practicing psychiatry in Spain. The sampling error was ±7.83%, under the assumption of an infinite study universe, with a maximum heterogeneity of –P = Q = 50%—and a confidence level of 95.5%. The frequency analysis was performed using Excel. 

## 3. Results

Firstly, the perception of 58.3% of psychiatrists was that venlafaxine was the most documented drug in the scientific literature in the last 5 years, whereas 27% thought that it was duloxetine, and 6.7% think that sertraline was the least documented drug. Secondly, the vast majority of physicians (99.4%) understood the mechanism of action of duloxetine as a SNRI. It was also universally known that it is approved for MDD. In addition, the vast majority of those surveyed knew that it is approved for anxiety and diabetic peripheral neuropathic pain.

In their opinion (99.4%), it presented advantages against other antidepressants. Specifically, in clinical situations such as musculoskeletal or non-specific pain (99.4%). A great proportion of those surveyed considered that duloxetine is also a good option for patients with cognitive impairment (29.5%) or dual pathology (27%). Very few would recommend it for the treatment of obsessive compulsive disorder (OCD) (4.9%). Almost all the participants (96%) agreed on 120 mg per day being the maximum dose for patients with MDD. The trade names of duloxetine that are better known by psychiatrists were oxitril (86%), cymbalta (86%), xeristar (80%), or dulotex (61%).

Regarding tolerance, 97% of the doctors surveyed alleged that their patients showed favorable outcomes (28% excellent tolerance, 69% good, 3% regular, and 0% bad). When doctors compared this drug with other SNRIs, 63% of them found that duloxetine is the drug that patients tolerate best, significantly more than desvenlafaxine, which 29% of them considered more tolerable. Besides, according to the perception of doctors, duloxetine was the drug preferred by patients (67%), while others mentioned fluoxetine or vortioxetine as preferred by patients (14 and 17%, respectively).

Eight out of ten of the physicians surveyed associated duloxetine with other psychoactive drugs: mirtazapine was the most common, followed by benzodiazepines and trazodone. Other psychoactive drugs mentioned in the open answer are indicated in Figure 1.

Duloxetine stands out from venlafaxine for its efficacy against pain. However, venlafaxine is somewhat more effective against sadness or apathy. See the complete diagram in Figure 2.

Finally, which drug psychiatrists would prescribe for particular clinical cases was revealed. Duloxetine was the most appropriate treatment for most kinds of patients analyzed: for a 63-year-old patient with diabetes and metabolic alterations (96% would recommend duloxetine); for a patient with anxious-depressive symptoms in the context of poorly controlled fibromyalgia (99%); and for patient with anxious-depressive symptoms in the context of tensional cephalea that did not ameliorate with amitriptyline (82%). The cases in which they would not always recommend this psychoactive drug were in males with attention deficit hyperactivity disorder (ADHD) and depressive disorders; 60% of the doctors surveyed would recommend duloxetine compared to 34% who would prescribe sertraline.

Duloxetine would be more frequently prescribed in women than in men, especially in cases of depressive episodes in menopause with urinary incontinence (70%) or with vasomotor symptoms (66%). Less than 20% would employ it for men with OCD and premature ejaculation. Almost 65% rejected it for elderly patients with depressive episodes and prostate problems. The probabilities for recommending duloxetine in different situations are detailed in Figure 3.

## 4. Discussion

In this work we have collected information around the use of duloxetine in comparison with other psychiatric drugs, and concerning psychiatrists’ knowledge and experience, along with patients’ preferences, symptoms, and well-being.

Firstly, among the antidepressants included in the questionnaire, duloxetine is the second highest ranking, with more indexed scientific publications (586) in PubMed according to a search performed at the end of December 2022 in which we applied filters “human” and “in the last 5 years”. Sertraline leads the rankings (822), closely followed by venlafaxine (570), while agomelatine has the lowest ranking (144). Physicians’ perception is that venlafaxine is the most investigated antidepressant in recent years, and that duloxetine is ranked in second place. Although both are close in terms of the number of publications, sertraline is actually in first place. Venlafaxine is approved by the FDA to treat and manage the symptoms of depression, social anxiety disorder, and cataplexy. However, it is also commonly used in other situations such as ADHD, fibromyalgia, diabetic neuropathy, complex pain syndromes, hot flashes, migraine prevention, post-traumatic stress disorder, obsessive compulsive disorder, and premenstrual dysphoric disorder [23]. Previous works have noticed that compared to duloxetine, venlafaxine has a 30-fold selectivity for serotonin reuptake. This could have important clinical consequences, as venlafaxine can be related to more notorious serotonergic adverse effects when compared with other SNRIs that present higher safety and reduced toxicity [14]. Hence, it might be possible that venlafaxine requires more research due to its higher number of adverse effects. Other plausible explanations could be that this drug was approved earlier by the FDA when compared to the others, or researchers feel that its potential uses may be worth further exploration [24].

On the other hand, the perception that duloxetine and venlafaxine are conceived as two antidepressants that generate a greater number of publications when compared to other types of antidepressants, such as sertraline, could be attributed to the fact that psychiatrists have different perceptions about SNRIs versus SSRIs based on their experience. Indeed, there is a huge scientific debate about what type of antidepressants could be more effective or safe, especially when comparing SSRIs or SNRIs. In this sense, previous works have demonstrated that SNRIs can exert more effective and faster antidepressant effects than other common medications used, such as serotonin selective reuptake inhibitors (SSRIs) [25], although these differences are still controversial according to the available literature [26,27]. Conversely, other works argue that sertraline is superior to other antidepressants in terms of efficacy (fluoxetine) or acceptability/tolerability (amitriptyline, imipramine, paroxetine, and mirtazapine) [28] and some recent reports shown that SSRIs such as sertraline are more commonly prescribed among depressive patients compared to other antidepressants such as SNRIs [29]. However, scientific studies comparing sertraline versus duloxetine or venlafaxine, report few differences between them in terms of improving the quality of life of depressed patients or reducing the severity of symptoms [30,31]. Hence, the clinician´s experience in this area may be a critical contributor to its effectiveness in individual patients and may indicate possible differences in the perception of SNRIs versus SSRIs.

Virtually all the interviewed psychiatrists noticed that duloxetine has certain benefits when compared to other antidepressants, especially in terms of ameliorating musculoskeletal or non-specific pain. Besides, they report excellent or good tolerability of its use in the majority of their patients, with maximum doses of 120 mg/day. Regarding the available literature, duloxetine has proven to be effective in the short- and long-term treatment of MDD, particularly for treating the core emotional symptoms, as well as the painful physical symptoms associated with depression [32]. The tolerability, safety, and efficacy of duloxetine is supported by previous systematic reviews at doses comprised between 60 and 120 mg/day, even in elderly patients or in those with concomitant illnesses [33,34]. In our work, physicians report that only 3% of patients show regular tolerability to duloxetine, whereas 28% and 69% display excellent and good tolerance, respectively. In agreement with this fact, previous works indicate that duloxetine is associated with few reported serious side effects, although there are some common adverse events consistent with the pharmacology of the molecule, mainly related to the gastrointestinal and the nervous systems [35]. Headaches, fatigue, diarrhea, nausea, and constipation are some common adverse effects of duloxetine use [13]. When compared to certain types of SSRIs, duloxetine may be associated with a higher rate of patients experiencing dry mouth, nausea, or insomnia, but less abdominal pain, and no differences in terms of diarrhea, urination problems, somnolence, constipation, or anxiety [36]. Thus, the clinicians’ perceptions are in accordance with the scientific literature, and this could explain patients’ preferential use of duloxetine when compared with other antidepressants.

When questioning physicians, duloxetine appeared to have advantages in its use for pain management, whereas venlafaxine was more effective in ameliorating sadness or apathy. According to a recent meta-analysis exploring the use of seven first-line antidepressants (fluoxetine, paroxetine, escitalopram, sertraline, fluvoxamine, venlafaxine, and duloxetine) both venlafaxine and duloxetine were the most effective drugs for the clinical management of MDD [16]. According to the scientific literature, duloxetine may be more effective in reducing anxiety and suicidal ideation in depressed patients [37] and also in terms of pain alleviation [38]. Simultaneously, some authors agree that venlafaxine could be a valid alternative for patients who do not tolerate or respond to SSRIs [39], whereas patients receiving duloxetine tend to have a more complex and costly antecedent clinical presentation [40]. Hence, it is likely that the perception that venlafaxine is superior to duloxetine in terms of ameliorating depressed mood or apathy could be related to which cases duloxetine is commonly prescribed for when compared to venlafaxine.

On the other hand, according to our results, duloxetine is frequently given with other psychoactive drugs. Mirtazapine is the most common, followed by benzodiazepines and trazodone. In this sense, there are some relevant works supporting the association of duloxetine with mirtazapine in patients with treatment-resistant depression [41], and some preclinical model studies endorse their synergic benefits to alleviate depression symptoms [42]. The combination of SNRIs with benzodiazepines seems to improve treatment outcomes in patients with comorbid anxiety and depression [43], whereas trazodone may be effective in depressed patients with comorbid insomnia, anxiety, or psychomotor agitation [44]. However, some authors are aware of the possible warnings associated with combining duloxetine with other antidepressants, in terms of possible toxicity and adverse effects [45]. The fact that duloxetine is commonly prescribed in combination with other psychotropic drugs can also be attributed to the more complex clinical presentations in which duloxetine is prescribed, as aforementioned.

We may find that duloxetine is a suitable treatment for MDD, diabetic neuropathy, fibromyalgia, and GAD. In our study, physicians would utilize it for patients with diabetes and metabolic alterations; however, not all of these complications have been studied. There are warnings in the literature about duloxetine-induced hepatotoxicity and it is not recommended for patients with chronic liver diseases. New advances in drug discovery and optimization have already demonstrated in vitro tests to reduce this hepatotoxicity [46]. On the other hand, duloxetine would be more frequently prescribed in women than in men. Despite the fact that this has been reported in previous works, it seems that the efficacy of duloxetine does not vary among both populations [47,48]. A plausible explanation of this fact is that the use of SSRI antidepressants is related to more severe sexual dysfunction in men [49], and also that duloxetine is related to less weight gain when compared to other antidepressants, which is a common adverse effect associated with the female gender [50,51]. Our results show that at least 7 out of 10 doctors would prescribe it for their female patients with depressive episodes in menopause and urinary incontinence. There is incipient evidence about duloxetine for women with stress urinary incontinence [52], validating the use of duloxetine by the psychiatrists interviewed in this work. Conversely, almost 65% reject it for elderly males with depressive episodes and prostate problems. This could be attributed to the fact that duloxetine may exert significant benefits in terms of ameliorating depressive symptoms and in the treatment of stress urinary incontinence after a prostatectomy, meaning adverse event rates are relatively high for these patients [53].

Finally, the preference of patients for duloxetine when it comes to musculoskeletal or unspecified pain in our results turns out to be supported by the scientific literature. In the last two decades, duloxetine has demonstrated a significant reduction in MDD-associated physical pain in double-blind placebo-controlled trials [54,55]. Neuropathic pain relief by duloxetine has also been reported for other comorbidities, with slight benefits in comparison to the use of other therapeutic alternatives [56,57]. It is also of note that some psychiatrists believe that duloxetine would not be indicated for elder patients with depressive symptoms and neuropathic diabetic pain. Despite some studies having found some benefits from duloxetine use when compared to other common drugs, such as gabapentin, the safety profile of the latter is superior to that of duloxetine, which may explain the results obtained [58].

The main strengths and relevance of our study lie in the participation of a significant number of psychiatrists from different hospitals and health centers located in various regions in Spain, relying on their clinical experience as professionals and on the translation of the scientific knowledge generated from bench to bedside. This allows us to access to critical information that would not be possible otherwise. However, our study also has some limitations. Firstly, we have only included psychiatrists from public health systems, so not all professionals in this field may be represented. Furthermore, the format of the questionnaire was based on closed questions, which may mean that the answers given are not as spontaneous as those that would be considered in a more informal environment or context. In the same way, in order not to take too much time from the professionals surveyed, we only included 20 different questions related to the use of duloxetine, leaving open the possibility of raising other relevant issues in the future.

## 5. Conclusions

According to the results obtained through the questionnaire, most psychiatrists consider that the use of duloxetine is well supported by scientific evidence. Furthermore, in their experience, duloxetine is quite well tolerated, even more so than other antidepressants. Interestingly, the results obtained in the present study show significant consistency with the available scientific literature regarding the use of duloxetine in MDD alone or presented with other comorbidities, such as fibromyalgia (Table 2), which, together with the fact that most psychiatrists understood its mechanism of action as well as the indications for which it is approved, can be considered an indicator of the adequate training of the psychiatrists surveyed.

## Figures and Tables

**Figure 1 brainsci-13-00333-f001:**
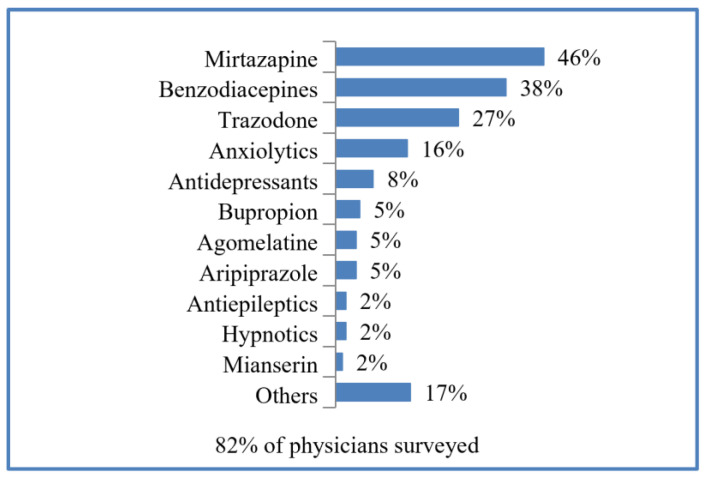
List of psychoactive drugs with which psychiatrists often associate duloxetine.

**Figure 2 brainsci-13-00333-f002:**
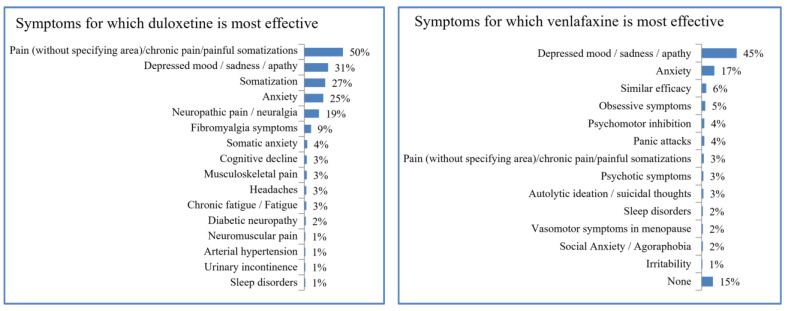
Comparison between duloxetine and venlafaxine and symptoms for which they are more effective.

**Figure 3 brainsci-13-00333-f003:**
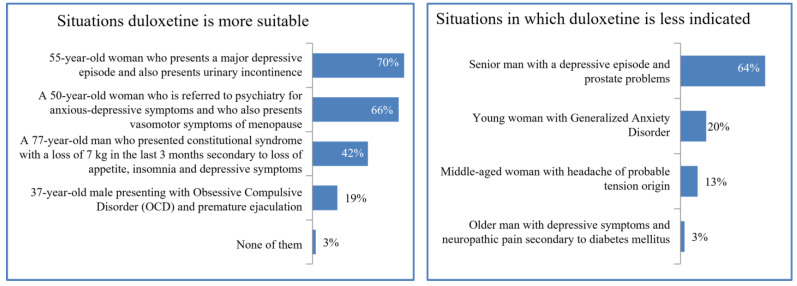
Situations for which duloxetine is the most and the least recommended antidepressant according to the doctors surveyed.

**Table 1 brainsci-13-00333-t001:** Questions included in the survey.

Questions Included in the Survey.	Answers
Among the following antidepressants, which do you think has generated the most indexed publications in the last five years? Order, please, from the first (the one that has generated the most posts) to the fourth (the one that has generated the fewest posts)	☐ Sertraline ☐ Agomelatine ☐ Duloxetine ☐ Venlafaxine
What is the mechanism of action of Duloxetine? (Single answer)	☐ Selective serotonin reuptake inhibitor ☐ Selective norepinephrine reuptake inhibitor ☐ Selective inhibitor of reuptake of norepinephrine and serotonin ☐ Selective norepinephrine and dopamine reuptake inhibitor ☐ I don’t know
In your opinion, does Duloxetine have specific advantages over other antidepressants? (Single answer)	☐ Yes ☐ No
In what clinical situations does Duloxetine have advantages over other antidepressants? (Multiple answer)	☐ Somatic symptoms (musculoskeletal, non-specific pain, etc.) ☐ Obsessive symptoms ☐ Dual pathology ☐ Elderly with cognitive impairment
How would you rate the tolerance of Duloxetine? (Single answer)	☐ Excellent ☐ Good ☐ Regular ☐ Bad
Can you please cite the commercial names of duloxetine that you remember? Please write down your answer.	
In a patient diagnosed with major depression, what would be the maximum dose of duloxetine that you would use? (Single answer)	☐ 30 mg/day ☐ 60 mg/day ☐ 90 mg/day ☐ 120 mg/day
In relation to subjective well-being, which antidepressant do you think patients prefer? (Single answer)	☐ Duloxetine ☐ Fluoxetine ☐ Agomelatine ☐ Vortioxetine
Do you usually associate Duloxetine with any other psychoactive drug? (Single answer)	☐ Yes ☐ No
With what type of psychotropic drugs do you most frequently associate Duloxetine? Please write down your answer.	
For which of the following diseases is Duloxetine approved? (Possible multiple answers)	☐ Major Depressive Disorder ☐ Diabetic peripheral neuropathic pain ☐ Fibromyalgia ☐ Generalized Anxiety Disorder ☐ Chronic musculoskeletal pain
In a 63-year-old patient with diabetes mellitus and metabolic disorders who is referred for the first time to psychiatry for presenting a major depressive episode, what antidepressant would you start the treatment with? (Single answer)	☐ Duloxetine ☐ Mirtazapine ☐ Trazodone ☐ Paroxetine
In which of the following clinical situations would you use duloxetine? (Possible multiple answer of the codes, 1,2,3 and 4)	☐ 55-year-old female presenting an episode major depressive disorder and also has urinary incontinence ☐ 50-year-old female referred to psychiatry for anxiety-depressive symptoms and who also presents symptoms vasomotors of menopause. ☐ 37-year-old male who presents a Disorder Obsessive Compulsive (OCD) and premature ejaculation ☐ 77-year-old male who presents syndrome constitutional with loss of 7 kg in the last 3 months secondary to loss of appetite, insomnia and depressive symptoms. ☐ In none of them
Which of the following antidepressants do you think is most indicated in a patient who is referred to psychiatry for presenting anxious-depressive symptoms in the context of poorly controlled Fibromyalgia? (Single answer)	☐ Vortioxetine ☐ Trazodone ☐ Sertraline ☐ Duloxetine
Which of the following antidepressants do you think is most indicated in a patient who is referred to psychiatry for presenting anxious-depressive symptoms in the context of a tension-type headache of months of evolution that has not improved with amitriptyline? (Single answer)	☐ Agomelatine ☐ Mirtazapine ☐ Duloxetine ☐ Sertraline
In your opinion, in which of the following clinical situations would Duloxetine be less indicated? (Single answer)	☐ Older man with a depressive episode and prostate problems ☐ Middle-aged woman with headache of probable tension origin ☐ Older man with depressive symptoms and neuropathic pain secondary to diabetes mellitus ☐ Young woman with Generalized Anxiety Disorder
Which of the following antidepressants do you think is most indicated in a 20-year-old male patient who is being followed up with Psychiatry for Attention Deficit Hyperactivity Disorder who suffers a depressive episode? (Single answer)	☐ Agomelatine ☐ Duloxetine ☐ Sertraline ☐ Trazodone
In what symptoms do you think Duloxetine is more effective than Venlafaxine? Please write down your answer.	
In what symptoms do you think Venlafaxine is more effective than duloxetine? Please write down your answer.	
In your opinion, which SNRIs (Serotonin and norepinephrine reuptake inhibitors) is better tolerated? (Single answer)	☐ Venlafaxine ☐ Venlafaxine retard ☐ Desvenlafaxine ☐ Duloxetine

**Table 2 brainsci-13-00333-t002:** A summary of the main results obtained in the questionnaire and the related scientific literature.

Result from the Survey	Scientific Literature	References
Duloxetine has advantages over other antidepressants, especially for patients with MDD and somatic manifestations (mainly pain).	Duloxetine has proven to be effective in the short- and long-term treatment of MDD, particularly for treating the core emotional symptoms as well as the painful physical symptoms related to depression.	[32,54,55]
The maximum dose of duloxetine for patients with MDD is 120 mg per day, with favorable responses in 97% of patients.	Duloxetine is safe, effective, and well-tolerated at doses comprised between 60 and 120 mg/day, even in elderly patients or in those with concomitant illnesses.	[33,34]
Compared to venlafaxine, duloxetine exerts greater efficacy against pain and somatization. However, venlafaxine is somewhat more effective against sadness or apathy.	Both venlafaxine and duloxetine are two of the most effective drugs for the clinical management of MDD.	[16]
	Duloxetine seems to be more effective in reducing anxiety and suicidal ideation in depressed patients and in pain alleviation. In general, patients receiving duloxetine tend to have a more complex and costly antecedent clinical presentation whereas venlafaxine could be a valid alternative in patients who do not tolerate or respond to SSRIs.	[37,38,39,40]
8 out of 10 clinicians prescribed duloxetine in combination with other agents, mainly mirtazapine, followed by benzodiazepines and trazodone.	Duloxetine is used with mirtazapine in patients with treatment-resistant depression, and some preclinical model studies endorse their synergic benefits to alleviate depression symptoms. The combination of SNRIs with benzodiazepines seem to improve treatment outcomes in patients with comorbid anxiety and depression, whereas trazodone may be effective in depressed patients with comorbid insomnia, anxiety, or psychomotor agitation. However, some authors are aware of the possible warnings related to combining duloxetine with other antidepressants in terms of possible toxicity and adverse effects.	[41,42,43,44,45]
Duloxetine is more frequently prescribed in women.	Duloxetine is more commonly prescribed to women, although the efficacy of duloxetine does not vary among male and female populations. Duloxetine is related to more severe sexual dysfunction in women when compared to men, and it is also related to less weight gain when compared to other antidepressants, a common adverse effect associated with the female gender.	[47,48,49,50,51]
7 out of 10 clinicians would recommend duloxetine in female patients with depressive episodes in menopause and urinary incontinence, although almost 65% reject it for elderly males with depressive episodes and prostate problems.	Duloxetine shows favorable effects in women with stress urinary incontinence, whereas in men with urinary incontinence and depressive symptoms after a prostatectomy the mean adverse event rates are relatively high.	[52,53]
Duloxetine can be recommended for alleviating musculoskeletal or unspecified pain, but not in elderly patients with depressive symptoms and neuropathic diabetic pain.	Neuropathic pain relief by duloxetine has also been reported in other comorbidities beyond MDD, with slight benefits in comparison to the use of other therapeutic alternatives, although its safety profile in neuropathic diabetic pain is limited when compared to other therapeutic alternatives.	[56,57,58]

## Data Availability

The data used to support the findings of the present study are available from the corresponding author upon request.

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
