# Peer review of "Current Opinions about the Use of Duloxetine: Results from a Survey Aimed at Psychiatrists"

_brainsci, 2023, doi:10.3390/brainsci13020333_

Round 1

Reviewer 1 Report

This article needs revision on following points.

Comments 1: The full name of abbreviations should be added in the first mentioned place.

Comments 2: Keywords should be alphabetical.

Comment 3: the incidence rates between 2015 to 50 2017 were 7.12, 7.35 and 8.02 per 1000 person-year respectively (Line 51). The underlined may be changed to person / year.

Comment 4: The article does not talk about the side effects of the drug duloxetine as compared to any other drug in use. A paragraph on side-effects and safety can be discussed.   

Comment 5: Why duloxetine is more commonly prescribed to women? Provide suitable references. 

Comment 6: Clinical data on use of duloxetine can be presented in the tabular form with suitable references.

Comments 7: The authors should take care of grammatical errors and spacing throughout the paper.

Comments 8: Please pay attention to the superscript and subscript needs revision in the whole Manuscript

Author Response

Reviewer 1:

This article needs revision on following points.

Comment 1: The full name of abbreviations should be added in the first mentioned place.

Response: We thank the reviewer for the comment. We have corrected the abbreviations that were missing throughout the manuscript.

Comment 2: Keywords should be alphabetical.

Response: We welcome the correction. We have already organized them alphabetically.

Comment 3: the incidence rates between 2015 to 50 2017 were 7.12, 7.35 and 8.02 per 1000 person-year respectively (Line 51). The underlined may be changed to person / year.

Response: We appreciate the recommendation. We have corrected it.

Comment 4: The article does not talk about the side effects of the drug duloxetine as compared to any other drug in use. A paragraph on side-effects and safety can be discussed.  

Response: Following the reviewer’s recommendations, we have added information about side-effects and safety in ‘Discussion’ section (lines 344-352).

Comment 5: Why duloxetine is more commonly prescribed to women? Provide suitable references.

Response: We agree with the reviewer about explaining more this fact. We added references in ‘Discussion’ section (lines 390-393). Some of the main reasons are that 1) SSRIs cause severe sexual dysfunction in women and 2) Specially in women, antidepressants like SSRIs are associated with greater weight gain when compared to duloxetine.

Comment 6: Clinical data on use of duloxetine can be presented in the tabular form with suitable references.

Response: We agree with the reviewer´s opinion of developing a table in order to compare our results with available literature. Hence, we have created table 2 in the section of conclusions.

Comment 7: The authors should take care of grammatical errors and spacing throughout the paper.

Response: We thank the reviewer for the comment. We have made proofreading of the whole manuscript and we have corrected the grammatical/spelling mistakes.

Comment 8: Please pay attention to the superscript and subscript needs revision in the whole Manuscript

Response: We welcome all the constructive comments made by the reviewer. We have made an extensive editing of the manuscript and the topic in order to make it more rigorous and complete.

Reviewer 2 Report

Dear Editor,

The article, entitled ‘Current opinions about the use of duloxetine: results from a 2 survey aimed to psychiatrists’ with manuscript number brainsci-2194753, explains the results of a scientific investigation on the employment of duloxetine and its possible superiorities over some other medications. There are some points listed below that may aid in strengthening the quality of the paper.

- Somatic symptoms in fibromyalgia, diabetes are different disease states (DS) in comparison to depression. So what is the aim of the study? If these DS are also topic of concern, no efficient result about them. Therefore abstract should be reorganized.

- In addition, drug repositioning should be emphasized in the introduction part, since the topic is duloxetine.

- Figure 1 should be reorganized. Antidepressants meaning does not make sense!!! One wants to see it in detail with each pharmacological name.

- Why to compare duloxetine specifically with venlafaxine?

- Conclusion is too short about the significance of the work.

- Ethical statement is absent (Legal permissions).

- The questionnaires followed are also absent!

Author Response

The article, entitled ‘Current opinions about the use of duloxetine: results from a survey aimed to psychiatrists’ with manuscript number brainsci-2194753, explains the results of a scientific investigation on the employment of duloxetine and its possible superiorities over some other medications. There are some points listed below that may aid in strengthening the quality of the paper.

Response: We appreciate the constructive comments made by the reviewer. We have made an extensive editing of the manuscript and the topic in order to make it more rigorous and complete.

- Somatic symptoms in fibromyalgia, diabetes are different disease states (DS) in comparison to depression. So what is the aim of the study? If these DS are also topic of concern, no efficient result about them. Therefore abstract should be reorganized.

Response: We thank the reviewer for the comment. The aim of this study was to deepen on the opinion and level of knowledge that psychiatrists have about duloxetine, and how and when do they use it. That is, we wanted to know what level of knowledge psychiatrist have about this antidepressant and what are the clinical situations in which they choose this treatment instead of other pharmacological options. We wanted to know all this through a survey of psychiatrists because it seemed to us the best way to obtain information from clinical practice. We have focused on Major Depressive Disorder (MDD), but we have included other pathologies in which the use of duloxetine is approved in order to a broader vision.

According to the reviewer´s recommendation, we have modified the abstract to make it clearer.

- In addition, drug repositioning should be emphasized in the introduction part, since the topic is duloxetine.

Response: We agree on reviewer’s suggestion about the topic. We modified the introduction regarding duloxetine (lines 63-89).

- Figure 1 should be reorganized. Antidepressants meaning does not make sense!!! One wants to see it in detail with each pharmacological name.

Response: Thanks for the suggestion. However, we have preferred to keep the names of the active pharmaceutical ingredient instead of the trade names because it seems more appropriate. Each active pharmaceutical ingredient has several trade names, and we have not asked psychiatrists for the different trade names of the different antidepressants. On the other hand, the figure would be very loaded, because instead of having one name for each drug, there would be several, which, in our opinion, would make it difficult to interpret the figure.

Although if the reviewer considers it essential that we change it, we will do so.

- Why to compare duloxetine specifically with venlafaxine?

Response: We appreciate the question raised by the reviewer. The reason is that both venlafaxine and duloxetine are the most representative and commonly prescribed SNRIs antidepressants. Therefore, we felt it appropriate to compare duloxetine with a similar drug that is well known and commonly prescribed.

- Conclusion is too short about the significance of the work.

Response: According to the reviewer´s suggestion, we have changed the conclusions section (427 – 435). In addition, we have included a paragraph with the strengths and limitations of our study (lines 414 – 425).

- Ethical statement is absent (Legal permissions).

Response: In this revised version of the manuscript, we have included the Ethical statement (lines 448-450).

- The questionnaires followed are also absent!

Response: In this revised version of the manuscript, we have included the questionnaire as Table 1. In the previous version we included it as supplementary material. Therefore, unless there has been an error, it has always been available to reviewers.
